# The Temperature-Dependent Retention of Introns in *GPI8* Transcripts Contributes to a Drooping and Fragile Shoot Phenotype in Rice

**DOI:** 10.3390/ijms21010299

**Published:** 2019-12-31

**Authors:** Bo Zhao, Yongyan Tang, Baocai Zhang, Pingzhi Wu, Meiru Li, Xinlan Xu, Guojiang Wu, Huawu Jiang, Yaping Chen

**Affiliations:** 1Key Laboratory of Plant Resources Conservation and Sustainable Utilization, South China Botanical Garden, Innovation Academy for Seed Design, Chinese Academy of Sciences, Guangzhou 510650, China; 1074854034@163.com (B.Z.); tangyongnian219@sina.com.cn (Y.T.); pzwu@scbg.ac.cn (P.W.); limr@scbg.ac.cn (M.L.); xxl@scbg.ac.cn (X.X.); wugj@scbg.ac.cn (G.W.); hwjiang@scbg.ac.cn (H.J.); 2University of Chinese Academy of Sciences, Beijing 100049, China; 3State Key Laboratory of Plant Genomics, Institute of Genetics and Developmental Biology, Chinese Academy of Sciences, Beijing 100101, China; bczhang@genetics.ac.cn

**Keywords:** cell division, cell wall, GPI8, GPI-APs, pre-mRNA splicing, rice (*Oryza sativa* L.)

## Abstract

Attachment of glycosylphosphatidylinositols (GPIs) to the C-termini of proteins is one of the most common posttranslational modifications in eukaryotic cells. GPI8/PIG-K is the catalytic subunit of the GPI transamidase complex catalyzing the transfer en bloc GPI to proteins. In this study, a T-DNA insertional mutant of rice with temperature-dependent drooping and fragile (*df*) shoots phenotype was isolated. The insertion site of the T-DNA fragment was 879 bp downstream of the stop codon of the *OsGPI8* gene, which caused introns retention in the gene transcripts, especially at higher temperatures. A complementation test confirmed that this change in the *OsGPI8* transcripts was responsible for the mutant phenotype. Compared to control plants, internodes of the *df* mutant showed a thinner shell with a reduced cell number in the transverse direction, and an inhomogeneous secondary wall layer in bundle sheath cells, while many sclerenchyma cells at the tops of the main veins of *df* leaves were shrunken and their walls were thinner. The *df* plants also displayed a major reduction in cellulose and lignin content in both culms and leaves. Our data indicate that GPI anchor proteins play important roles in biosynthesis and accumulation of cell wall material, cell shape, and cell division in rice.

## 1. Introduction

Attachment of glycosylphosphatidylinositol (GPI) moieties to protein C-termini is one of the most common posttranslational modifications in eukaryotic cells [1]. Proteins modified by GPI attachment are called GPI anchor proteins (GPI-APs). The GPIs have a common backbone whose structure is EtNP-6-mannose (Man)-1,2-Man-1,6-Man-1,4-glucosamine(GlcA)-1,6-inositol-P-lipid [2]. The mature GPIs are transferred en bloc to proteins by GPI transamidase (GPI-T) complexes in the ER. The GPI-T complex consists of five subunits: PIG-K/GPI8 (human/yeast orthologs), PIG-S/GPI17, PIG-U/GAB1, PIG-T/GPI16, and GPAA1/GAA1 [3]. GPI8 and GPAA1 are the catalytic subunits of GPI-T [3]. GPAA1 catalyzes the formation of a bond between the substrate protein’s omega-site and the GPI lipid anchor’s phosphoethanolamine [4]. GPI8 has a dual role: it carries out the proteolytic cleavage of the C-terminal signal sequence of the precursor protein, then forms an amide bond between the protein and the ethanolamine phosphate of the GPI [5].

Loss of GPI anchoring in plants results in lethality, while a reduction in GPI seriously affects plant growth and development. Glucosamine-6-P acetyltransferase (GNA) is involved in de novo biosynthesis of UDP-*N*-acetylglucosamine (UDP-GlcNAc). The transfer of GlcNAc from UDP-GlcNAc to phosphatidylinositol is the initial reaction carried out by the GPI assembly [6]. Mutation of GNA in rice (*gna1*, [7]) affects cell shape and the growth of roots under low temperature conditions, and in Arabidopsis (*lig*; [8]) it causes temperature-dependent growth defects and ectopic lignin deposition. *SETH1* and *SETH2* encode Arabidopsis homologs of two conserved proteins involved in the first step of the GPI biosynthetic pathway. The *seth1* and *seth2* mutations block pollen germination and tube growth [9]. In the *aptg1* mutant, pollen tubes are able to elongate but suffer a reduction in their micropylar guidance capacity and there is embryo lethality in homozygotes [10]. The *PNT1* gene encodes the Arabidopsis homolog of mammalian *PIG-M*. The *pnt1* mutants are seedling lethal [11]. Loss of function of *AtGPI8* (*atgpi8-2*) in Arabidopsis is lethal. A mutation in *atgpi8-1* that results in low enzyme activity disrupts growth, fertility, and formation of stomata [12].

A number of GPI-APs have been identified and/or predicted in plants [13]. The functions of several GPI-APs have been revealed based on the study of mutants. Most of the reported functions of GPI-APs are associated with cell wall biosynthesis and maintenance. COBRA (COB), the first GPI-AP to be found in Arabidopsis, is involved in cell wall synthesis. Mutations in the *COB* gene of Arabidopsis reduce the content of crystalline cellulose in cell walls in the root growth zone [14]. Loss of function of the fasciclin-like arabinogalactan-protein (FLA) SOS5 causes decreased root growth and thinner cell walls [15]. Another GPI-AP, AtSHV3, contains two glycerol phosphate choline diester phosphodiesterase domains. Plants doubly mutate for this gene and its homolog *SVL1* shows shorter root hairs, swollen guard cells, abnormal hypocotyls, reduced cellulose content, ectopic lignin deposits, and fragile cell walls, indicating that *SHV3* and its homolog participate in primary cell wall synthesis [16]. Fourteen FLAs in Arabidopsis have been identified as GPI-APs. Analysis of the *fla11*/*fla12* double mutant suggested that Arabidopsis FLA11 and FLA12 may enhance plant stem strength and stem modulus of elasticity [17]. Mutations in *AtSKU5* (which contains a cupredoxin-like domain) change the growth properties of roots and influence the direction of root growth and cell expansion [18]. In several mutants, loss of function of GPI-AP severely affects pollen development, pollen germination efficiency, and pollen tube growth. Examples include classical AGPs [19], aspartic protease [20], ENODL11-15 proteins that contain an early-nodulin like/plastocyanin domain [21], and LORELEI [22]. Other functions of the GPI-APs include immune recognition, transmembrane signaling, embryogenesis, formation of stomata, and export of cuticular wax [23].

In rice, COB-like proteins and CLD1/SRL1 are the only GPI-APs whose functions have been studied in detail. *BC1* belongs to the COB-like family which is expressed mainly in developing sclerenchyma cells and vascular bundles. *BC1* mutations cause not only a reduction in cell wall thickness and cellulose content but also an increase in lignin [24]. T-DNA insertion in *OsBC1L4* results in abnormal cell expansion, dwarfing, and fewer tillers in rice [25]. The *CLD1/SRL1* gene encodes a GPI-anchored membrane protein that modulates leaf rolling and other aspects of rice growth and development [26,27].

In the present study, we further elucidated the biology function of GPI-APs in rice plants using a T-DNA insertional mutant that shows a drooping and fragile (*df*) phenotype. Tail-PCR and DNA sequencing indicated that the T-DNA was inserted downstream of the stop codon of *OsGPI8*, the ortholog of the Arabidopsis *GPI8* gene. Insertion of the T-DNA causes defective intron splicing of the *OsGPI8* pre-mRNA, especially under higher temperature conditions. Intron retention in transcripts of the *OsGPI8* gene results in the premature stop codon. Results from the anatomical analysis showed that the insertion of mutation led to a decreased cell number in the culm and a large change in cell wall structure. Reduced cellulose and lignin contents were deposited in the mutant culms and leaves correspondingly. Our results provide new insight into the function of GPI-anchored proteins in plant cell division in rice plants.

## 2. Results

### 2.1. Isolation of the Drooping and Fragile Shoot Rice Mutant

In a previous study, we constructed transgenic rice plants with RNA interference of the rice red chlorophyll catabolite reductase 1 gene (*RCCR1i*) and analyzed their phenotype. The *RCCR1i* transgenic plants displayed lesion mimic spots in older leaves and these leaves died off early [28]. Among the homozygous *RCCR1i* lines grown in the greenhouse (day/night, 30–36 °C/25–30 °C), one line exhibiting drooping shoots (Figure 1A,B), fragile culms, and leaves (Figure 1C,D) was identified, and we named this rice mutant *df* (drooping and fragile). About 1/4 (8/30) of the plants of the T2 generation were drooping and fragile, indicating that a single recessive nuclear locus was likely to be responsible for this mutant phenotype. The mechanical strengths of *df* culms and leaves were found to be reduced to, respectively, ca. 17% and 39% for those of WT and *RCCR1i* lines when their breaking forces were measured (Figure 1E,F). The shoots of *df* seedlings showed the same morphology as those of *RCCR1i* seedlings under lower temperatures (LT; 19–23 °C), while manifesting the drooping phenotype at higher temperatures (HT; 29–33 °C) (Appendix A). The mutant phenotype of *df* was therefore inferred to be temperature-dependent.

### 2.2. Changes in Morphological and Anatomical Structures of Culms

The leaves and culms of *RCCR1i* plants had the same mechanical strength as those of wild-type plants: one normal *RCCR1i* line was therefore chosen as the control for the next test. The *df* plants were shorter than *RCCR1i* plants (Figure 1B). Having measured the internode length in four internode-containing stems, we found that the lengths of the first top internode and the second top internode of *df* plants were decreased by about 22.6% compared to *RCCR1i* plants (Figure 2A,B). The culm diameter of *df* plants was also significantly less than that of *RCCR1i* plants (Figure 2C). Upon examining transverse sections of the internodes of mature stage plants, we observed that the shell of the culm is thinner (Figure 2D), and the length of the vascular bundle is shorter (Appendix A) in *df* plants than in *RCCR1i* plants. No significant difference was observed in the average size of parenchyma cells in culms between *df* plants and *RCCR1i* plants, but the parenchyma cell layer number and peripheral sclerenchyma thickness in *df* plants were significantly reduced compared to those in *RCCR1i* plants (Figure 2E; Appendix A). These results suggested a reduction in cross-sectional cell number in *df* internodes. To determine the pattern of deposition of cell wall polymers, ultra-thin sections of *RCCR1i* and *df* bundle sheath cells were examined by transmission electron microscopy. The secondary wall layer of *RCCR1i* had a uniform thickness and appearance, while showing a clearly layered structure with different electron-dense materials in *df* cell walls (Figure 2F).

### 2.3. Changes in the Morphological and Anatomical Structures of Leaves

Compared to those of *RCCR1i* plants, 11–28% of mature leaves in *df* plants were shorter (Appendix A). No differences in the morphology and arrangement of epidermal cells and stomata were observed in the fully expanded leaves of *df* (Appendix A). To explore the causes of the reduced mechanical strength of *df* leaves, we observed the cell shape and sclerenchyma structure in the main vein by means of transverse sectioning of leaves at the tillering stage. Light microscopy revealed that cells lying between the phloem and epidermis at the distal end of the main vein were shrunken in *df* leaves. The sclerenchyma cells of the main vein displayed greater intracellular transparency in *df* than in *RCCR1i* (Appendix A). Electron microscopy indicated that some of the parenchyma and sclerenchyma cells within the distal end of the main vein in *df* leaves were shrunken, and that the cell walls of these sclerenchyma cells were significantly thinner in *df* than in RCCR1i leaves (Figure 3). The mean thickness of the radial walls of two adjacent sclerenchyma cells within the outermost layer and the secondary outer layer in the distal end of the main vein in *df* was 1.24 ± 0.34 μm and 1.37 ± 0.47 μm, respectively, while the corresponding values were 2.19 ± 0.54 μm and 3.13 ± 0.58 μm in *RCCR1i* plants, according to statistical analysis of thirty cells.

### 2.4. Differences in Cell Wall Composition

The reduction in mechanical strength and defects in wall structure observed in *df* suggested that the cell wall composition may be altered in the mutant plants. Because cellulose and lignin are the major components of the second cell walls influencing mechanical strength in plants, we therefore analyzed these components in culms and leaves. Cellulose content was reduced by ca. 27% in culms and by ca. 32% in leaves in *df* plants compared to *RCCR1i* plants (Table 1). The *df* plants also showed a decrease in lignin content in both leaves and culms. By contrast, most of the neutral sugars from culms and leaves in *df* plants were increased.

### 2.5. Cloning of the Gene

Southern blot analysis using a hygromycin phosphotransferase gene fragment as a probe revealed that the *df* mutant contained only one copy of the T-DNA insertion (Figure 4A). The flanking sequences bordering the T-DNA tag were obtained using the thermal asymmetric interlaced (TAIL)-PCR method [29]. The insertion site was identified as being 879 bp downstream of the stop codon of the LOC_Os02g12740 (Os02g0219400) gene, and 7575 bp before the start codon of the LOC_Os02g12730 (Os02g0219200) gene (Figure 4B). The LOC_Os02g12730 gene encodes a β-galactosidase family protein (OsBGAL1), which is homologous to Arabidopsis BGAL1 (At3g13750). The LOC_Os02g12740 gene (OsGPI8) encodes a putative catalytic subunit of GPI-T, which is the homolog of Arabidopsis GPI8 (At1g08750).

To identify the candidate gene, we examined the levels of expression of *OsBGAL1* and *OsGPI8* genes in shoots of control (*RCCR1i*, *DF*), heterozygote (*DF*/*df*), and homozygote (*df*) plants by semi-quantitative PCR. A primer pair hybridizing to regions located in exons 9 and 11 was used to test the expression of the *OsBGAL1* gene. The results showed that the *OsBGAL1* gene was weakly expressed in shoots of control plants (*DF*), but was more highly expressed in both heterozygote and *df* plants (Figure 4C). Being a recessive mutant, the *OsBGAL1* gene may not be responsible for the *df* phenotype. A primer pair hybridizing to regions located in exons 4 and 5 was used to test the expression of the *OsGPI8* gene (Figure 4B). The expected product, 266 bp in length, of the *OsGPI8* gene amplified using this primer pair was observed in *DF* plants, whereas a larger fragment was obtained in *df* plants. The heterozygote (*DF*/*df*) plants had both bands in the PCR products (Figure 4C). Results from the sequencing of these products indicated that the larger fragment from *df* plants contains the intron in the cDNA fragment. Subsequently, cDNA containing the complete coding sequence was amplified from *DF* and *df* seedlings. After cloning and sequencing, the PCR product from *DF* plants was found to be the predicted 1315 bp in length, while it was 3780 bp in length from *df* plants, retaining all the introns of the gene (Figure 4D). No DNA sequence variation was detected in the 3780 bp fragment from *df* plants. We therefore deduced that defective pre-mRNA processing of the *OsGPI8* gene transcript may be responsible for the mutant phenotype of *df* plants.

The *GPI8* gene is found ubiquitously across eukaryotic organisms and its sequence is highly conserved. Homology analysis of OsGPI8 revealed that the protein has a high amino acid sequence similarity to *Saccharomyces cerevisiae* GPI8 (46%) and *Arabidopsis thaliana* GPI8 (72%). The catalytic active site of GPI8 (His-166 and Cys-208) is conserved (Appendix A). The coding region of the *OsGPI8* gene is 3603 bp in length, including 9 introns and 10 exons, which encodes 404 amino acids (Figure 4B, Appendix A). While the intron retention in transcripts of the *OsGPI8* gene results in the premature stop codon and the encoded truncated protein lacking the main active domain (Appendix A). Therefore, the retained introns in the DF transcript make *OsGPI8* lose functions.

To confirm whether the *OsGPI8* gene was responsible for the *df* mutant phenotype, the cDNA containing the complete coding sequence of the wild-type *OsGPI8* gene was cloned into the binary vector pCAMBIA2301. *OsGPI8* driven by the cauliflower mosaic virus 35S promoter (OsGPI8-Oe) was then introduced into *df* mutant plants via *Agrobacterium tumefaciens*-mediated transformation. A total of 11 transgenic lines (*df GPI8-Oe*) were obtained. Of these, two independent T2 lines that displayed a ca. 3:1 segregation ratio (17:5 and 7:2) of normal to *df* mutant plants were studied in detail (Figure 5A). The presence of the artificial *OsGPI8* in genomes of transgenic plants was determined by PCR analysis. An introduced cDNA fragment of the expected length (266 bp) was found in all morphologically normal plants, but it was absent from plants with the mutant phenotype in both of the two independent T2 lines (Figure 5B). In addition, complemented plants (*df GPI8-Oe*) showed similar cellulose content (292.5 ± 1.61 to 300.7 ± 1.12 mg per gram of alcohol-insoluble residues) in leaves with *RCCR1i* plants. These results confirmed that overexpression of the wild-type *OsGPI8* cDNA complemented the mutant phenotype. The *OsGPI8* gene was expressed in all tissues tested by real-time quantitative PCR (qRT-PCR), including roots, stems, leaves, panicles, and seeds (Figure 5C), suggesting that it plays roles in most tissues of rice plants.

A previous report has shown that GPI protein affects stomatal clustering and callose content in Arabidopsis seedlings [12]. We therefore determined the callose contents in *RCCR1i* and *df* leaves at the four-leaf stage, and observed that *df* leaves had a significantly higher callose content (43.58 ± 0.24 μg g^−1^ fresh weight (FW)) than *RCCR1i* leaves did (21.26 ± 0.37 μg g^−1^ FW), but no stomatal clustering was observed in *df* leaf blades like *atgpi8-1* mutant (Appendix A) [12].

### 2.6. Temperature Influences the Pre-mRNA Processing of OsGPI8 Transcripts in df Plants

The mutant phenotype of the *df* was found to be temperature-dependent. To investigate whether temperature influenced the pre-mRNA processing of *OsGPI8* transcripts, transcripts of the *OsGPI8* gene were analyzed by RT-PCR using total RNA extracted from shoots of 14-day seedlings grown at higher temperatures (HT; 29–33 °C) and at lower temperatures (LT; 19–23 °C). For this analysis, three primer pairs were designed for the PCR amplification, with the intended products corresponding to different exon-encoded regions of the 5′, middle, and 3′ regions of the *OsGPI8* transcripts (Figure 6A). We found that the band corresponding to normally spliced transcripts for *OsGPI8* was weak when *df* shoots were grown at HT, whereas it was strong in RNA from *df* seedlings grown under LT conditions (Figure 6B). Most introns were retained in the transcripts (Figure 6B), and only a low level of normally spliced transcripts containing the complete coding domain sequences (Figure 6C) was observed in *df* seedlings grown at HT. These results indicate that temperature did affect the relative proportions of normal mRNA and intron-retaining forms of mRNA in *OsGPI8* transcripts from *df* mutant plants, and the temperature-dependent *df* phenotype may result from reduced pre-mRNA splicing of *OsGPI8* under HT conditions. On the other hand, no transcriptional silencing of full length or partial regions of the *OsGPI8* pre-mRNA was performed when plants were grown at HT.

### 2.7. Change in Length of 3′-UTR of OsGPI8 Transcripts

Intron retention and alternative splicing are associated with polyadenylation sites in plants [31]. The *OsGPI8* gene contains at least two putative polyadenylation sites (Appendix A). Normally, the 3′-UTR of *OsGPI8* is about 257 bp in length according to the results of our 3′-RACE analysis and also the EST data in GenBank (https://www.ncbi.nlm.nih.gov/), but a 3′-UTR fragment over 1000 bp in the *OsGPI8* transcript was detected in the *df* plants. To test the abundance of 3′-UTR with different lengths in *RCCR1i* and *df* plants, two primer pairs whose locations were indicated in Figure 6A were designed. qRT1 amplified by the primer pair 1F and 1R represents both the short and long 3′UTR, while qRT2 amplified by the primer pair 2F and 2R only represents the long 3′UTR. The expression levels of total *OsGPI8* transcripts had no difference between *RCCR1i* and *df* plants under either HT or LT conditions (Figure 6D), but the abundance of *OsGPI8* transcripts with the longer 3′-UTR in the *RCCR1i* plants was around 1.7% of that in *df* plants under HT conditions (Figure 6E). Compared to HT conditions, the amount of *OsGPI8* transcripts with the longer 3′-UTR was reduced by ca. 63% in *df* plants under LT conditions (Figure 6E). These results indicated that the length of *OsGPI8* transcripts increased at the 3′-end, especially under HT conditions, in *df* plants.

## 3. Discussion

The GPI anchor biosynthesis pathway has been studied intensively in yeast and in humans, and it is believed that this pathway may also be conserved in plants. In Arabidopsis, homologous genes exist for almost all the known components of the GPI anchoring pathway, although currently the only genes that have been studied are *SETH1, SETH2, PEANUT1 (PNT1), APTG1,* and *AtGPI8* [9,10,11,12]. These genes encode enzymes that catalyze specific steps in the biosynthesis of the anchor. The yeast and mammalian GPI transamidase complexes consist of five subunits [3]. Like Arabidopsis [12], rice has homologs of all five subunits, with all but one subunit encoded by only one gene: they are LOC_Os02g12740, which is a homolog of At1g08750 and Gpi8/PIG-K, LOC_Os01g48980 (At5g19130, Gaa1p/GAA1), LOC_Os01g67960 (At3g07180, Gpi17p/PIG-S), LOC_Os11g28980 (At3g07140, Gpi16p/PIG-T), LOC_Os02g46350, and LOC_Os01g64990 (At1g12730 and At3g27325, Gab1p/PIG-U).

Alternative splicing, which can lead to intron retention in transcripts, is widely observed in plants [32,33]. Changes in splicing are induced by environmental stimuli including ambient temperature [34,35]. There is an interplay between the messenger RNA processing reactions, splicing, and polyadenylation, which take place co-transcriptionally [36]. Intron retention and alternative splicing are associated with polyadenylation sites in plants [31]. Defects in pre-mRNA splicing of *OsGPI8* transcripts at higher temperatures in *df* plants is an interesting but as yet inexplicable phenomenon. The *OsGPI8* gene contains at least two polyadenylation sites (Appendix A). RT-PCR analysis indicated that when *df* plants were grown under HT conditions they had the same amount of OsGPI8 transcripts as at LT, but most of them had a longer 3′-UTR and retained the majority of the nine introns (Figure 6). These results imply that the usage of alternative polyadenylation sites was one factor leading to intron retention in *OsGPI8* gene transcripts in the *df* plants. The mechanism underlying intron retention in OsGPI8 transcripts after the insertion of the T-DNA in *df* plants may be that, on the one hand, the basic terminators of rice GPI8 have variability, and the insertion of foreign DNA disrupts the termination message, which results in a large increase in the length of 3′-UTR at a higher temperature. On the other hand, the DNA changes in structure or chromatin accessibility [37], or transcriptional interference [38] by the foreign *GUS* gene when it is inserted near the *OsGPI8* gene, and this merits further study. Expression of *OsGPI8* cDNA containing the complete coding sequences was found to rescue the rice *df* mutant (Figure 5A,B), confirming that it is the T-DNA insertion and defective pre-mRNA splicing of the *OsGPI8* transcript that cause the drooping and fragile phenotype of this *df* mutant.

Our study of the *df* mutant allowed us to explore the role of GPI-APs in morphology, growth, and development in rice. By analysis of the rice protein database of 66,338 proteins (Phytozome, http://www.phytozome.net/index.ph-p), a total of 481 proteins were predicted to be GPI-APs and were identified by two or more methods (shown in Appendix A). Eight members of BC1-like family proteins were predicted as GPI-APs. The *CWA1/BC1* gene encodes a COBRA-like protein and functions in the regulation of secondary cell wall deposition and assembly in rice [24,39,40]. Mutation of *BC1* causes a reduction in cellulose content but an increase in lignin content [24]. A decrease in cellulose content but an increase in pectin and starch contents were identified in *Osbc1l4* mutants that showed abnormal cell expansion, dwarfing, and fewer tillers phenotype [25]. OsBC1L5 is required for pollen tube elongation in rice [41]. Except for the BC1 family, *CLD1/SRL1* also encodes a *GPI*-anchored membrane protein that modulates leaf rolling and other aspects of rice growth [26,27]. The *cld1* mutant exhibits significant decreases in cellulose and lignin contents in cell walls of leaves.

According to annotation of rice GPI-APs, they are mainly involved in Glycosyl hydrolases family 17 (60), plastocyanin-like family proteins (41), lipid transfer like proteins (LTPs, 36), and Fasciclin-like (23). Glycosyl hydrolases were suggested to degrade the cell wall polymer, and are required for cell wall organization during tissue morphogenesis [42]. The plastocyanin-like family proteins are plant-specific blue copper proteins that were reported to regulate lignin biosynthesis induced by oxidative stress [43]. The LTPs may function in cuticular wax export [23] as well as cell wall extension [44]. Fasciclin-like GPI-APs were reported to play roles in regulating cell-type specification, cellulose deposition, and formation of secondary cell wall composition in Arabidopsis. In summary, many GPI-APs function on the formation of cell walls in plants. They can regulate the biosynthesis of cell wall composition such as cellulose and lignin, cellulose assembly, and the formation of the microfibril crystallinity in cell walls. Here, the *df* plants have a reduction of cellulose and lignin contents (Table 1), suggesting GPI-APs may positive regulate the biosynthesis and accumulation of cellulose and lignin in rice. Moreover, we speculate that plastocyanin-like family proteins might be involved in the pathway of OsGPI8 regulating affecting cell wall components since plastocyanin-like proteins directly affect lignin biosynthesis. The dense structure of the cell wall was changed in the *df* mutant (Figure 2F and Figure 3), indicating the roles of GPI-APs in regulating the deposition and assembly of cell wall polymers during cell wall formation in rice.

GPI-APs were found to modulate cell division in Arabidopsis. AtAGP19 is a lysine-rich, classical AGP that is predicted to be attached to the plasma membrane by a GPI anchor. The *atagp19* mutant has a reduced number of abaxial epidermal cells in rosette leaves, indicating a decrease in cell division [45]. The reduction in cell division among parenchyma cells in internodes of *df* plants (Figure 2D,E) indicated that, in rice, GPI-APs have a function related to cell division. The homology protein of AtAGP19 was not found in the rice protein database. The cell division related to GPI-APs in rice remains to be explored. The fragile culm and leaves phenotype of *df* plants is likely to be due to the reduction in parenchymal cell layers and in the deposition and assembly of secondary cell wall materials (Figure 2E and Figure 3D). The drooping leaves may result from the significant reduction in wall thickness and cellulose content in sclerenchyma cells, which are thus insufficiently strong to support the blade.

The loss of GPI anchoring is lethal in Arabidopsis [10,11]. A T-DNA insertion line of *AtGPI8*, *atgpi8-2* is distributed as a heterozygous line, and it does not segregate out homozygous plants. In addition, *atgpi8-1/atgpi8-2* plants were found to be severely dwarfed; they never flowered, and did not survive into maturity [12]. The *atgpi8-1* mutant has a mutation consisting of a G to A substitution at base pair 125 which results in the replacement of Arg-42 with Gln-42 [12]. While the *atgpi8-1* mutation strongly disrupts plant growth, it is not lethal. Since it is not a null mutant of *GPI8*, it is understandable that the *df* mutation is not lethal even when plants are grown at HT. No stomatal clustering [12] in leaf blades or male defects [9,10,11] were observed in *df* plants, but there was a reduction in plant height and in cell number in the culm shell (Figure 1 and Figure 2). These results suggest distinct physiological and developmental differences between Arabidopsis and rice in response to a lower level of GPI anchoring.

GPI-APs are widespread in eukaryotic cells. However, direct evidence for the role of GPI-APs in rice physiology and development has hitherto been largely lacking. In conclusion, in this work, we confirm that the T-DNA insert the downstream of the stop codon in the *OsGPI8* gene in rice, which results in intron retention and the knockdown of the *GPI8* gene under higher temperature conditions. Analysis of knockdown of the *GPI8* gene demonstrates the importance of GPI-APs in the accumulation of cell wall material, cell expansion, and cell division, as well as the morphology and mechanical strength of rice plants.

## 4. Materials and Methods

### 4.1. Plant Materials and Growth Conditions

The *df* mutant was generated from transgenic *RCCR1i* T-DNA insertion lines of the japonica rice cv. Huazhiwu. Wild-type rice (*Oryza sativa* L.), *OsRCCR1i*, and *df* plants were grown in a greenhouse under natural daylight at South China Botanical Garden, CAS, in Guangzhou during the natural growing seasons. For temperature treatments, the seedlings were cultivated in rice nutrient solution [46] in square basins and placed in growth chambers (16 h light/8 h dark, higher temperature (29–33 °C), lower temperature (19–23 °C)).

### 4.2. Cloning of the df Gene and Complementation Test

DNA fragments flanking the T-DNA insertions were amplified from the mutant plant genomic DNA using TAIL-PCR [29]. The PCR products were cloned into a pMD18-T vector (Takara; http://www.takara.co.jp/english) and sequenced by a company (Augct, Beijing, China). BLAST analysis indicates that the gene obtained from the *df* mutant was on chromosome 2. The *OsGPI8* cDNA was cloned into a pMD18-T vector by reverse transcription-PCR using primers of fGPI8F and fGPI8R (shown in Appendix A). For the complementation test, the *OsGPI8* cDNA clone with the complete CDS was digested with *Xba* I and subcloned into the corresponding site of pCAMBIA2301 under the control of the cauliflower mosaic virus (CaMV) 35S promoter. The PCR primers used are listed in Appendix A. Agrobacterium strain EHA105 harboring the above vector was used to transform mutant calli via *Agrobacterium tumefaciens*-mediated transformation [7].

### 4.3. RNA Isolation and Expression Analysis

Total RNA was isolated from different tissues using Trizol reagent (Invitrogen, Carlsbad, CA, USA) following the manufacturer’s instructions. First-strand cDNA was synthesized from 3 μg of the total RNA using M-MLV reverse transcriptase (Promega, Madison, WI, USA) and was used as a template for semi-quantitative PCR analysis after normalization relative to a rice polyubiquitin1 gene (*OsRUB1*, LOC_Os06g46770) [8]. Primers used for this part of the study are listed in Appendix A. Quantitative real-time PCR was performed using an ABI 7500 fast system with a SYBR Premix Ex Taq^TM^ II (perfect real-time) kit (Takara) and the following program: predenaturation at 95 °C for 30 s, followed by 40 cycles of 95 °C for 5 s, 60 °C for 34 s, and a final dissociation cycle of 95 °C for 15 s, 60 °C for 1 min, and 95 °C for 15 s. Each PCR assay was run in duplicate for three independent biological repeats. The comparative Ct method [30] was used to determine the relative expression of the target genes, with *OsRUB1* as the reference gene.

### 4.4. Southern Blot Analysis

Approximately 5 µg of DNA from each sample was digested with *Eco*R I, *Eco*R V, *Hind* III, and *Sac* I (Takara) and separated on a 0.8% (*w*/*v*) agarose gel. The restriction enzyme recognition sites occur only once each within the T-DNA, allowing the transgene copy number to be estimated. The fractionated genomic DNA was transferred to a positively charged nylon membrane using 20× saline sodium citrate (SSC; 1.5 M NaCl, 0.15 M sodium citrate) (Amersham Pharmacia Biotech, Piscataway, NJ, USA). Probes were prepared using a PCR DIG probe synthesis kit (Roche, Basel, Switzerland). Hybridization was performed according to the DIG Application Manual (Roche). Hybridization was at 42 °C and washing was performed under high-stringency conditions at 65 °C.

### 4.5. Mechanical Strength Test

The breaking forces for rice culms or leaves were measured with a universal force/length testing device (model Zwick Z2010, Ulm, Germany). For the measurements, second internodes (with leaf sheath) and flag leaf blades of WT, RCCR1i and *df* plants cut to the same length were used.

### 4.6. Microscopic Analysis

For light microscopic analysis, second internodes excised from rice plants were fixed in an FAA solution (formaldehyde 5 mL, glacial acetic acid 5 mL, and 70% ethanol 90 mL). The samples were dehydrated through an ethanol gradient and embedded in paraffin. For cell wall staining, sections were cut at a thickness of 10 μm with a microtome (Leica RM2255, Leica Biosystems, Buffalo Grove, IL, USA) and stained in 0.1% methylene blue for 3 to 5 min. The stained sections were examined and details recorded by taking pictures under a light microscope (ZEISS, Axio Imager A2, Jena, Germany) with a color CCD camera (AxioCam MRc). For electron microscopy, internodes excised from rice plants were fixed in 2.5% glutaraldehyde and 2% paraformaldehyde, then dehydrated through an ethanol series and embedded in Spurr resin. Ultra-thin sections were made using glass knives on an Ultracut E ultramicrotome (Leica). Ultra-thin sections (0.1 μm) were stained with 2% uranyl acetate for 1 h and 6% lead citrate for 20 min and observed with an electron microscope (JEM-1010, Jeol; http://www.jeol.com).

For microscopic analysis of leaf blade samples, the third leaves from four-leaf stage seedlings were fixed in 2.5% glutaraldehyde, vacuum treated, and post-fixed in 2% (*w*/*v*) osmium tetroxide, and washed with 0.1 M phosphate-buffered saline several times until osmium tetroxide was eliminated. Subsequently, samples were dehydrated successively with 30% (4 °C, 20 min), 50% (4 °C, 20 min), 70% (4 °C, overnight), 80% (RT, 15 min), 90% (RT, 15 min), 100% alcohol (RT, 30 min for twice), and finally propylene oxide (RT, 30 min for twice). After dehydration, samples were treated by different ratios of propylene oxide and epoxy resin and then embedded in pure epoxy resin. Semithin sections (2 μm) were stained with toluidine blue for 30 s, then observed and recorded by taking pictures under a light microscope with a color CCD camera (Zeiss). Ultra-thin sections were stained with 2% uranyl acetate for 1 h and 6% lead citrate for 20 min and observed with an electron microscope (JEM-1010, Jeol; http://www.jeol.com).

### 4.7. 3′-RACE PCR

The 3′-Race PCR was performed using a BD SMART™ RACE cDNA Amplification kit, referring to the manufacturer’s instructions. The primers are listed in Appendix A.

### 4.8. Cell Wall Component Measurement

Carbohydrate was isolated and assayed according to the methods described previously [47,48,49]. The samples (the second internode and flag leaf blades) were dried and ground into a fine powder using a food processor (JYL-B020). The powder was washed three times in phosphate buffer (50 mM, pH 7.2), extracted twice with 70% ethanol at 70 °C for 1 h, and dried under vacuum to produce alcohol-insoluble residue (AIR). De-starching was performed by treating AIR with pullulanase M3 (0.5 U mg^−1^, Megazyme, http://www.megazyme.com) and α-amylase (0.75 U mg^−1^, Sigma) in 0.1 m NaOAc buffer (pH 5.0) overnight. The destarched AIRs further hydrolyzed into monosaccharides by Trifluoroacetic acid (TFA) and were reduced with sodiumborohydride. The alditol acetate derivatives produced by acetic anhydride treatment were subjected to GC-MS according to Zhang and Zhou (2017) [49]. The remains of TFA treatment were hydrolyzed with sulfuric acid. After the development of color with anthrone in concentrated sulfuric acid, the absorption of each reaction at 625 nm was measured on a spectrophotometer [49,50].

The lignin content was measured using the acetyl bromide assay [51]. About 10 mg of AIR for each sample was dissolved in 0.6 mL of 25% acetyl bromide in glacial acetic acid and heated at 50 °C for 2 h with occasional mixing. After cooling, the samples were centrifuged (3000× *g*, 15 min) and the supernatants were used for lignin quantification with hydroxylamine hydrochloride solution. Lignin (471003, SIGMA-ALDRICH) was used as the standard. The absorbance at 280 nm for each sample was measured using a spectrophotometer (UV-1800, MAPADA).

The callose content was measured as described previously [52] with some modifications. About 100 mg of fresh leaves were ground into a fine powder in liquid nitrogen after washing 3 times with ethanol to eliminate autofluorescence and then suspended in 1 mL 1 M NaOH. Samples were incubated at 80 °C for 15 min to solubilize the callose. The mixture was centrifuged (10,000× *g*, 15 min) and the supernatants were used for the callose assay. For callose determination, 0.1% (w/v) aniline blue was used for the reaction and 1,3-β-glucan (Pachyman, Megazyme) was used as the standard. Fluorescence of the assay mixture was read in a spectrofluorometer (PE LS55, excitation 400 nm; emission, 510 nm; slit width, 10 nm).

### 4.9. Statistical Analysis

Three to six biological repeats were used for all experiments, and all data in this work were analyzed with a Duncan test [53] using the SAS software package (http://www.sas.com/en_us/software/sas9.html).

## Figures and Tables

**Figure 1 ijms-21-00299-f001:**
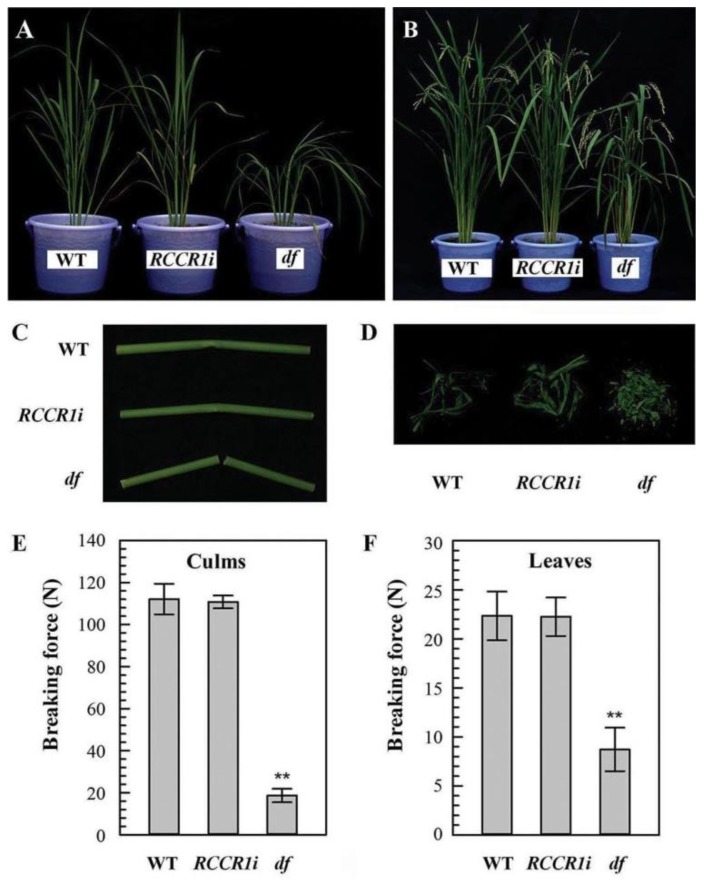
The phenotypes of *df* plants. (**A**) Tillering stage plants. (**B**) Filling stage plants. Brittleness of culms (**C**) and leaves (**D**). Breaking force for culms (**E**) and leaves (**F**). The second internode and flag leaves of filling stage plants were used for the test. When testing the culm, the leaf sheath was included. Error bar indicates ± SD from at least four biological repeats. ** Indicates significant difference (*p* < 0.01) between the *df* (drooping and fragile) mutant and *RCCR1i* plants.

**Figure 2 ijms-21-00299-f002:**
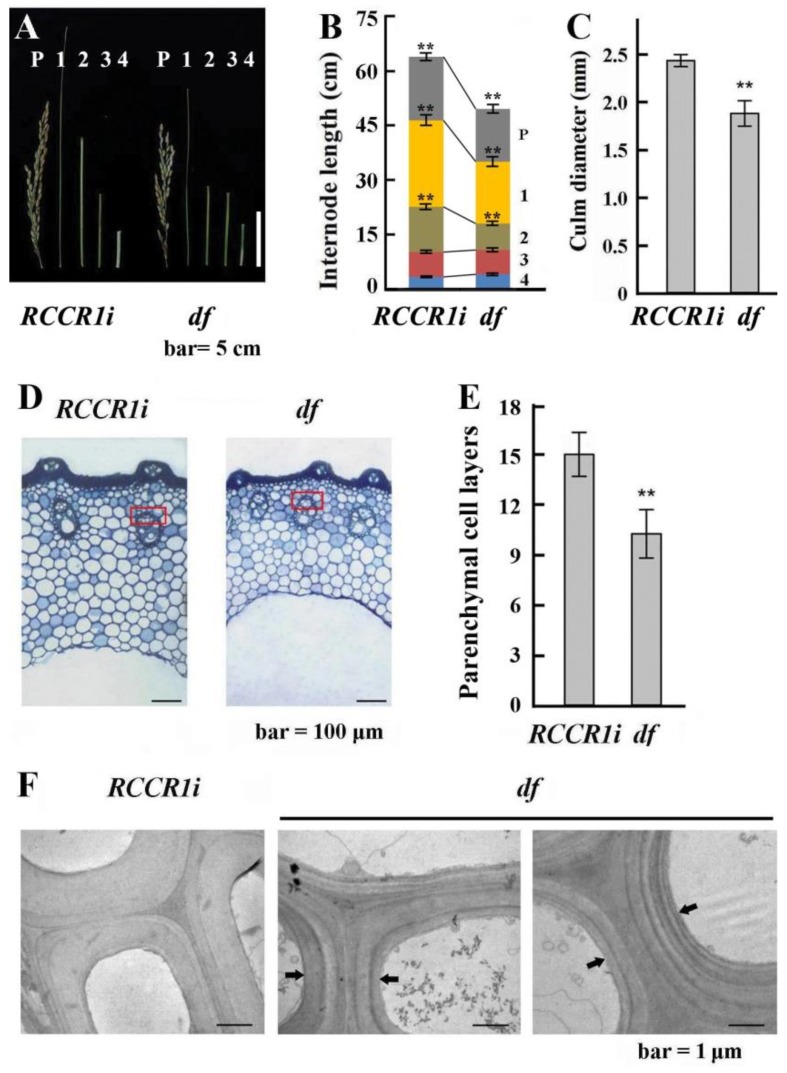
Differences in height and anatomical structures of internodes in *RCCRi* and *df* plants. (**A**) Morphology of the internodes. Each internodes are numbered below the panicle (P) to the base. (**B**) Statistical analysis of length changes among different internodes between *RCCRi* and *df* culms. (**C**) Statistical analysis of culm diameter changes of the second internodes between *RCCRi* and *df* plants. Error bar in (**B**) and (**C**) indicates ± SD from 15 plants. (**D**) Cellular analysis of *RCCRi* and *df* culms. Transverse sections of culms in the second internodes showing the reduced cell layers in *df* culms. (**E**) Statistical analysis of numbers of parenchyma cell layers in the second internodes of *RCCRi* and *df* culms. Error bar indicates ± SD from six measurements. ** Significant difference (*p* < 0.01). (**F**) Ultrastructure of cell walls in vascular bundle sheath cells of the region indicated by the rectangles in (**D**). Arrows indicate regions contain layers of different electron-dense materials in *df* culms.

**Figure 3 ijms-21-00299-f003:**
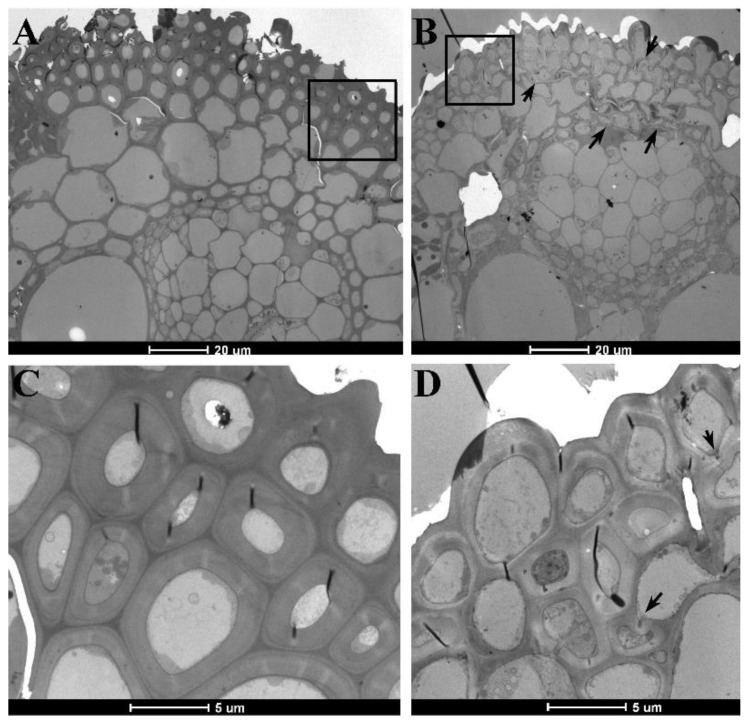
Electron microscopic observation of the sclerenchyma in the main leaf vein. Top of the main vein in leaves of *RCCR1i* (**A**) and *df* (**B**) plants. (**C**,**D**) Magnified sections of (**A**,**B**) showing the irregular shape and thin walls of sclerenchyma cells in the main leaf vein of *df* leaves. Arrows indicate the irregular shape cells.

**Figure 4 ijms-21-00299-f004:**
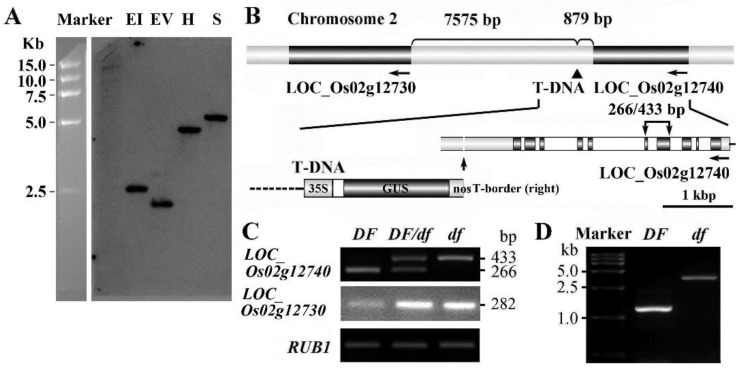
Cloning of the *df* gene. (**A**) Southern blot for the *df* mutant using a *Hygromycin* probe. DNA was digested with *Eco*R I (EI), *Eco*R V (EV), *Hind* III (H), and *Sac* I (S). DNA size markers are indicated on the left. (**B**) Location of the T-DNA insertion. The T-DNA is located between the LOC_Os02g12730 and LOC_Os02g12740 genes. For the LOC_Os02g12740 gene, dark bars represent its ten exons. (**C**) Differences in expression of LOC_Os02g12730 and LOC_Os02g12740 among *df* mutant, heterozygote (*DF*/*df*), and *RCCR1i* (*DF*), determined by RT-PCR analysis. Localization of the binding sites in exons 4 and 5 for the primer pair used for the testing of LOC_Os02g12740 transcripts is indicated in (**B**). Lengths (bp) corresponding to the predicted cDNA and genomic DNA are indicated. (**D**) RT-PCR of the transcripts containing the complete coding sequence of the *OsGPI8* gene.

**Figure 5 ijms-21-00299-f005:**
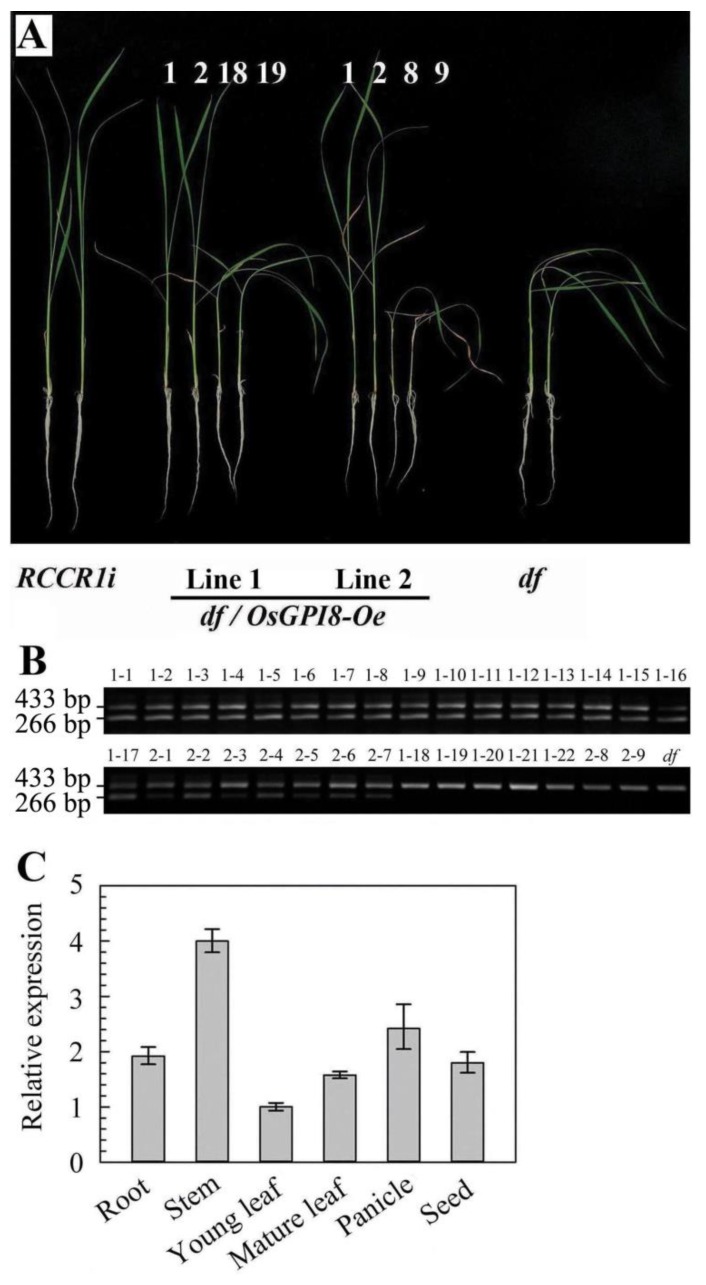
Complementation of the *df* mutant by the *OsGPI8* gene. (**A**) Two-week-old seedlings of two representative T2 lines of *df* plants overexpressing *OsGPI8* (*df*/*OsGPI8-Oe*) grown under 29–33 °C conditions. Phenotype segregation (the ratio of normal phenotype to *df* phenotype is 17:5 and 7:2, in two independent lines, respectively) was observed among two independent *df*/*OsGPI8-Oe* T2 families. (**B**) Genomic PCR analysis of *df*/*OsGPI8-Oe* T2 families. The 433 bp bands are products amplified from the rice genomic DNA of the *OsGPI8* gene, while the 266 bp bands are products from the artificial *OsGPI8* gene (which contains no intron). The 266 bp bands are absent in these plants. The binding sites for the primer pair are indicated in Figure 2B. (**C**) Analysis of expression of the *OsGPI8* gene. Each PCR assay was run in duplicate for three independent biological repeats. The comparative Ct method [30] (Livak and Schmittgen, 2001) was used to determine the relative expression of the target genes, with *OsRUB1* as the reference gene.

**Figure 6 ijms-21-00299-f006:**
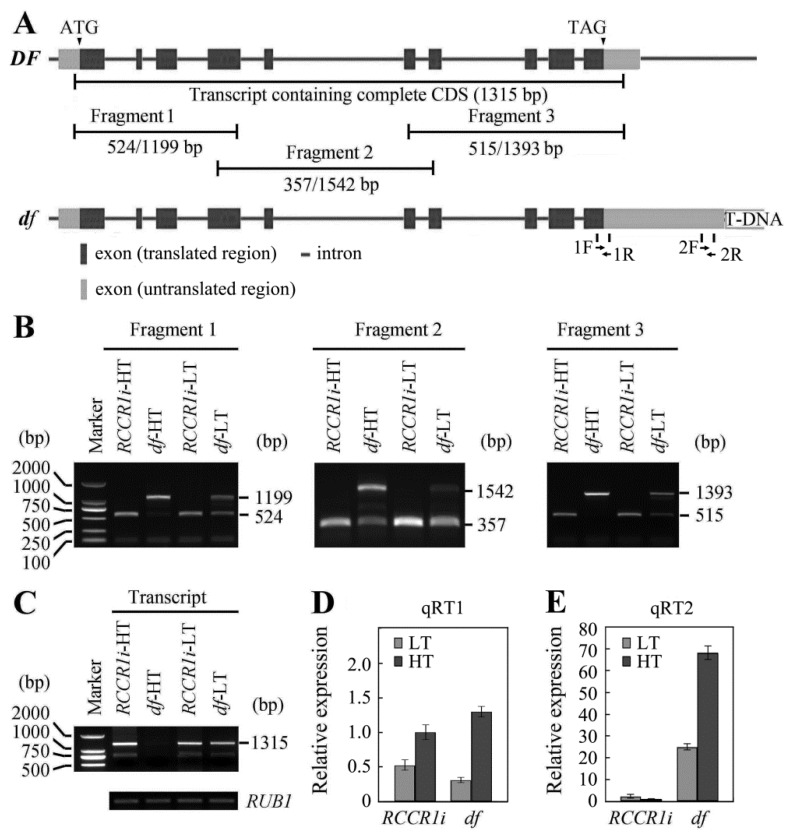
Temperature-dependent pre-mRNA splicing in *df* plants. (**A**) Distribution of primer pairs used for (q) RT-PCR along the OsGPI8 gene. The lengths (bp) of predicted cDNAs/genomic DNAs are indicated. (**B**) RT-PCR products of different regions of *OsGPI8* transcripts. Bands corresponding to predicted cDNAs and genomic DNA are indicated. (**C**) RT-PCR products for transcripts which contain the complete coding domain sequence. Bands corresponding to predicted cDNAs are indicated. (**D**) qRT-PCR analysis of total OsGPI8 transcripts using the primer pair of 1F and 1R indicated in (A) in shoots grown at different temperatures. (**E**) qRT-PCR analysis of OsGPI8 transcripts with the longer 3′-UTR using the primer pair of 2F and 2R indicated in (A) in shoots grown at different temperatures. Relative expression was normalized to that of the reference gene OsRUB1 (internal control). Bars show standard deviations of the repeats. Each assay was run in duplicate for three independent biological repeats. HT = higher temperature (29–33 °C). LT = lower temperature (19–23 °C).

**Table 1 ijms-21-00299-t001:** Measurement of the cell wall composition in *df* and *RCCR1i* plants.

Compound (μg mg^−1^ AIR)	Culms *RCCR1i*	*RCCR1i* *df*	Leaf Blades *RCCR1i*	Leaf Blades *df*
Cellulose	453.74 ± 8.13	331.9 3 ± 4.79 **	326.91 ± 5.62	221.21 ± 7.06 **
Lignin	142.41 ± 3.84	104.45 ± 1.01 **	98.72 ± 0.99	85.54 ± 1.42 **
**Neutral Sugars**				
Rhamnose	2.15 ± 0.08	2.44 ± 0.05 *	2.76 ± 0.09	2.97 ± 0.08
Fucose	1.20 ± 0.02	1.18 ± 0.02	1.41 ± 0.04	1.51 ± 0.01
Arabinose	32.58 ± 0.70	34.97 ± 0.65 *	41.79 ± 0.88	45.40 ± 0.79 *
Xylose	212.59 ± 3.74	317.89 ± 5.00 **	219.57 ± 4.83	229.71 ± 3.58 *
Mannose	2.19 ± 0.06	2.16 ± 0.04	2.56 ± 0.07	2.79 ± 0.03
Galactose	24.24 ± 0.296	19.89 ± 0.27 **	17.10 ± 0.30	18.56 ± 0.09 **
Glucose	135.75 ± 0.83	87.60 ± 1.58 **	27.86 ± 0.52	26.11 ± 0.31

*/** Significant difference compared with *RCCR1i* (*p* < 0.05/0.01).

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
