# Peer review of "The Temperature-Dependent Retention of Introns in GPI8 Transcripts Contributes to a Drooping and Fragile Shoot Phenotype in Rice"

_ijms, 2019, doi:10.3390/ijms21010299_

Round 1
Reviewer 1 Report
In my opinion, the aim of the work is not fully clear. This point should be highlighted in comparison with what has been achieved in previous studies. Additionally, no functional studies or mutational approaches were performed. These points should be also clarified with explanation.
1. Introduction
The aim of the paper should be more clarified and highlighted in this section.
2. Results
Figure 1: Do you mean SD or mean ± SD? Please clarify this point.
2.2. Changes in Morphological and Anatomical Structures of Culms
Figure 2: All figures of statistical analyses are not understandable, where the relation between X-axis and Y-axis is not described below the figure.
2.3. Changes in the Morphological and Anatomical Structures of Leaves
Figure 3. The description below the figure is very confusing and difficult to understand. The figure should be reorganized with more clarification.
2.4. Differences in Cell Wall Composition
In my opinion, the authors did not test all the cell wall composition, where they focused mainly on the sugar content (carbohydrate). This point should be clarified, and the importance of the chosen components should be also clarified in the text.
Table 1. The data in Table 1 are not clear and the table should be reorganized.
3. Discussion
This section needs more improvement, where the discussion of the obtained results are poorly discussed and should be rationalized with more adequate previous studies.
4. Materials and methods
4.3. RNA Isolation and Expression Analysis
Lines 388-389. Based on which criteria the primers have been chosen? This point needs clarification in the text.
4.8. Cell Wall Component Measurement
I would like to see the all chromatogram data of the analyzed silylated sugar by GC-MS. This could be added in the supplementary file.
Finally, I recommend the authors to check the whole paper for grammatical and typing errors as well as the list of abbreviations. For instance, some abbreviations are not listed such as SD (standard deviation).
Author Response
Comments and Suggestions for Authors
Reviewer 1
In my opinion, the aim of the work is not fully clear. This point should be highlighted in comparison with what has been achieved in previous studies. Additionally, no functional studies or mutational approaches were performed. These points should be also clarified with explanation.
Response: Thank you for your nice comments. We highlighted the significance of our results in the Introduction and made a detailed comparison in the discussion as well in the revised manuscript. Furthermore, we added a supplementary data about the annotation of all the GPI-APs from rice plant, which helps us to understand the significance of our results and also the function of GPI-Aps better.
Introduction
The aim of the paper should be more clarified and highlighted in this section.
Response: Thank you for your constructive comments. We rewrote the last paragraph of introduction and clarified the significance of our results. New paragraph was written as following: “In the present study, we further elucidated the biology function of GPI-APs in rice plants using a T-DNA insertional mutant that shows a drooping and fragile (df) phenotype. Tail-PCR and DNA sequencing indicated that the T-DNA was inserted downstream of the stop codon of OsGPI8, the ortholog of the Arabidopsis GPI8 gene. Insertion of the T-DNA causes defective intron splicing of the OsGPI8 pre-mRNA, especially under higher temperature conditions. Intron retention in transcripts of the OsGPI8 gene results in the premature stop codon. Results from anatomical analysis showed that the insertion mutation led to a decreased cell number in the culm and a large change in cell wall structure. Reduced cellulose and lignin contents were produced in the mutant culms and leaves correspondingly. Our results provide a new insight into the function of GPI-anchored proteins in plant cell division in rice plants”.
Results
Figure 1: Do you mean SD or mean ± SD? Please clarify this point.
Response: Thank you for the comments. SD should be ± SD in our manuscript and we have already clarified this point.
2.2. Changes in Morphological and Anatomical Structures of Culms
Figure 2: All figures of statistical analyses are not understandable, where the relation between X-axis and Y-axis is not described below the figure.
Response: Thank you for the comments. We have already added detailed descriptions of X-axis and Y-axis below figures in the revised manuscript with "Track Changes".
2.3. Changes in the Morphological and Anatomical Structures of Leaves
Figure 3. The description below the figure is very confusing and difficult to understand. The figure should be reorganized with more clarification.
Response: Thank you for the comments. We have already reorganized the description in figure 3 in the revised manuscript with "Track Changes".
2.4. Differences in Cell Wall Composition
In my opinion, the authors did not test all the cell wall composition, where they focused mainly on the sugar content (carbohydrate). This point should be clarified, and the importance of the chosen components should be also clarified in the text.
Response: Thank you for your comments. Cellulose, hemi-cellulose and lignin that are related to mechanical force were determined. We used xylose content to represent hemi-cellulose in the previous version. Now we added the content of hemi-cellulose in the revised manuscript and all the revised parts were used "Track Changes".
Table 1. The data in Table 1 are not clear and the table should be reorganized.
Response: Thank you for your comments. We have already reorganized Table 1 in the revised manuscript.
Discussion
This section needs more improvement, where the discussion of the obtained results are poorly discussed and should be rationalized with more adequate previous studies.
Response: Thank you for your nice comments. We reorganized the Discussion. And we added a supplementary data about the annotations of all the GPI-APs from rice plant, which helps us to understand the significance of our results and also the function of GPI-APs better.
Materials and methods
4.3. RNA Isolation and Expression Analysis
Lines 388-389. Based on which criteria the primers have been chosen? This point needs clarification in the text.
Response: Rice polyubiquitin1 (OsRub1) is expressed constitutively in rice plant. We added the Locus number and also reference in the revised manuscript.
4.8. Cell Wall Component Measurement
I would like to see the all chromatogram data of the analyzed silylated sugar by GC-MS. This could be added in the supplementary file.
Response: We only detected several specific cell wall components with the method we mentioned in the article, and all the data we obtained have been already presented in the manuscript. Now we corrected the Method and added the reference in the revised manuscript. What we rewrote is “The destarched AIR are further hydrolyzed into monosaccharides by trifluoroacetic acid (TFA) and reduced with sodium borohydride. The alditol acetate derivatives produced by acetic anhydride treatment are subjected to GC-MS according to Zhang and Zhou (2017) [32].”
Finally, I recommend the authors to check the whole paper for grammatical and typing errors as well as the list of abbreviations. For instance, some abbreviations are not listed such as SD (standard deviation).
Response: We checked and revised the manuscript carefully based on your nice comments. And we also reorganized the list of abbreviations in the revised article.
Reviewer 2
Overall, I enjoyed reading this manuscript. It is well written with only a few minor typographical / English expression errors (see below under Minor points for details). However, there are a few areas that require further detail and/or clarification.
Line 95 – can the number of T2 plants (x of y plants with the drooping and fragile phenotype) be provided?
Response: Thank you for your comments.1/4 plants show the drooping and fragile phenotype, and we added the sentence “eight of thirty plants in the T2 generation showed drooping and fragile phenotype” in the revised manuscript with “Track Changes”.
Line 117 – the authors described the vascular bundles of the df mutant culm as shorter but that they have no difference in width (Fig. S1). Is this due to less cells being within the vascular bundle (as is observed in the parenchyma layer) and/or a reduction in expansion of these cells? Please mention that there is no effect on the width of the vascular bundle within the body of the text.
Response: Thank you for your comments. In FigS1, df culm showed the shorter length but normal width in vascular bundles, therefore, we said the vascular bundles of the df mutant culm was shorter. For the avoidance of confusion, we revised the sentence as “the length of the vascular bundle is shorter’ in the revised manuscript with “Track Changes”.
Figure 2 – Where do the cells shown in Fig. 2F sit in relation to Fig. 2D? Some labels should be added to Fig.’s 2D and 2F to indicate cell types and contents, respectively. Are the authors sure that the cell walls of the df mutant are electron-dense? The cellular contents of the displayed cells have condensed contents suggesting that the fixation of the plant material wasn’t optimal, hence, the differences may be artefactual.
Response: Thank you for your comments. Yes, the electron-dense could be caused in the process of the fixation. But here we would like to show the different layers in the cell wall that was indicated with arrows in Fig 2F. Now we changed the sentence as “Whilst showed a clearly layered structure with different electron-dense materials in df cell walls” in the revised article with “Track Changes”.
Line 140-141, 143-144 – What is meant by the “abaxial end of the main vein”. The term abaxial refers to the bottom leaf blade surface. Do the authors mean the distal or proximal end of the main vein? Please clarify.
Response: Thank you so much for your suggestion. Yes, it should be the distal/proximal end of the main vein. We revised the article accordingly.
Figure 3 – please label sections with the different cell types
Response: Thank you for your comments. Figure 3C and 3D are the close-up of sections from A and B, so we labeled the sections and also the different cell types in the revised Figure 3 and its legend as well.
Table 1 – the values within this table do not appear in columns. Please provide units at the top of the table. Can the authors please comment on whether any (1,3;1,4)-β-glucan is present? Why are there only two callose values? To which samples do they belong to?
Response: We corrected the format of Table 1, added the units at the top of the table based on your nice comments. In the revised Table, we thought it’s better for us to put the callose value in the section of “2.5. Cloning of the Gene”, instead of Table 1. Therefore, the description in the revised manuscript is as following: Mutations in the AtGPI8 gene cause the atgpi8-1 mutant form stomatal clustering and accumulate higher levels of callose in Arabidopsis seedlings [12]. We determined the callose (1,3-β-glucan) contents in RCCR1i and df leaves at the four-leaf stage, and observed that df leaves had a significantly higher callose content (43.58±0.24 μg g-1 fresh weight (FW)) than RCCR1i leaves (21.26±0.37 μg g-1FW). But no stomatal clustering was observed in df leaf blades as described above (Supplementary Figure S3B).
Line 196-197 – The 282 bp band appears to be absent from the DF plant rather than weakly expressed according to the included image. Can a more convincing image be provided? It is difficult to judge if the OsBGAL1 is highly expressed or not in this context. A change in wording to “more highly expressed” is suggested.
Response: We used an improved image in Figure 4 in the revised article and changed the sentence based on your reasonable suggestion.
Line 223 – the number of T2 lines with a ~3:1 segregation ratio be provided? The reviewer recommends that the high temperature growth conditions be mentioned within the text in addition to the legend to Fig. 5 legend as these are the conditions under which a phenotype should be observed.
Response: We added the separation ratio and the description of phenotype in the revised manuscript according to your nice comments. And we also added the sentence “The ratio of normal phenotype to df mutant phenotype stands at 17: 5 and 7: 2 in two independent lines, respectively” in the legend of Figure 5.
Figure 5 – the bands in Fig. 5B lanes 1-1 and 2-1 are difficult to see? Can a more convincing image be added instead? It would be useful to point out that lines 1-18, 1-19 and 2-8 and 2-9 are "nulls" for the introduced DF cDNA, at least in the legend.
Response: Fig. 5B was replaced by an improved image and the missing band in lines 1-18, 1-19 and 2-8 and 2-9 was indicated in the Figure 5.
Figure 6 – What are the black and white arrows within the green boxes? Is the length of the last green box indicative of the shorter 257 bp 3' UTR? It is unclear where the 2 polyA tail addition sites are located in this figure and in Figure S5B (refer to Line 287). Presumably the location of qRT1 and qRT2 primer pairs detect the different transcripts but it is not clear from the diagram. Please clarify.
Response: We re-organized the Figure 6 in the revised manuscript and made all the data more clear according to your nice suggestions. PolyA tail was indicated in Figure S5B.
Line 318 – insert after be, "one or a combination of the following possibilities."
Line 360 – add in the citation to reference 38
Response: We added the reference in the corresponding position and reorganized the order of references as well.
Line 375 – list the name of the DNA sequencing company
Response: We added the name of the company in the revised paper.
Line 388 – please provide a locus number for the rice RUB1 gene
Response: We added the locus number LOC_Os06g46770 in the revised paper.
Line 399 – list the type of nylon and the manusfacturer
Response: We added the name of the company in the revised paper.
Line 411, 416, and 423 – please add more details of the drying and embedding process
Response: We added the details about the drying and embedding process in the revised manuscript as “Subsequently, samples were dehydrated successively with 30% (4ºC, 20 min), 50% (4ºC, 20 min), 70% (4ºC, overnight), 80% (RT, 15 min), 90% (RT, 15 min), 100% alcohol (RT, 30 min for twice) and finally propylene oxide (RT, 30 min for twice). After dehydration, samples were treated by different ratio of propylene oxide and epoxy resin, and then embedded in in pure epoxy resin.”
Line 306-325 - The early part of the Discussion focusses on the possible mechanism/s by which the introns of the premRNA transcript of DF aren’t spliced. Was genetic complementation also attempted with DF gDNA to confirm the importance of the native DF locus context to this nonsplicing?
Response: Thank you for your suggestions. Here we used the full length of cDNA (including 5’UTR, ORF and 3’UTR) to make genetic complementation in df background in our research.
Supplemental material
Line 40 – Replace “Statistic analysis of the toppest three leaves length” with “Statistical analysis of the length of the top three leaves”
Response: We replaced the sentence based on your nice comment with “Track Changes”.
Line 42 – what do the arrows indicate? Please add cell labels to the image
Response: Arrows indicated a column of stomatal cells that do not differ between df and RCCR1i plants. We added this sentence in the legend of Fig. S3.
Line 46-47 – What do the authors mean by “The experiment comes from triplicate for 4 independent biological repeats.” Are the images within Fig. S4 taken from one biological replicate and is representative of the others? Could the main anatomical features of the sections be labelled?
Response: We repeated this experiments for three times and used 4 independent plants for each experiment, since we needed to exclude that the shrinkage structure in df mutant was caused by the fixation of plant materials. Now we reorganized the sentence in the legend of Fig.S4 and labelled the main difference between df and RCCR1i.
Figure S5 – it is unclear where the two polyadenylation sites are in the DF locus. Please mark.
Response: We marked the polyadenylation sites in the image as 3907 and 4409, and added the detailed description about these sites in the legend of Figure S5 accordingly.
The major omission to the manuscript is the lack of description as to the consequence of nonspliced introns within the DF transcript at the end of the Results section. What is the impact on the DF protein when the introns aren't spliced out? Is the mature protein truncated due to a premature stop codon or does it encode a different C-terminus compared to the wild type protein? Is it predicted to still be functionally active? An understanding of the impact on the pool of GPI-anchored proteins would be helpful to gain an understanding of the phenotype of this mutant. This would then set the biological context of these findings within the discussion. Can the authors provide insight as to why so many different cell wall components differ between the RCCR1i lines and the df mutant?
Response: Thank you for your nice comments. The mature protein truncated is due to a premature stop codon in the transcript. It lacks the main active domain and therefore the functions of OsGPI8. We added some sentences in the “2.5. Cloning of the Gene” to describe the consequence of the retained introns in DF transcript as “While the intron retention in transcripts of the OsGPI8 gene results in the premature stop codon and the encoded truncated protein lacking the main active domain. Therefore, the retained introns in DF transcript makes OsGPI8 lost functions.”
We annotated 481 GPI-anchored proteins from rice (Supplementary Table 2) and discussed the pool in the “Discussion”. From the annotation in Supplementary Table 2, we can see that GPI-anchored proteins play multiple roles in the Cell wall deposition and therefore the cell wall components in df are remarkably different from the RCCR1i.
Minor points:
Line 34 – describe EtNP in full prior to using this abbreviation.
Response: The abbreviation list has been completed in the revised version according to reviewers’ nice comments.
Line 63 – should read root hairs not root hair
Response: We changed “root hair” to root hairs in the revised article.
Line 72 – should read functions not function
Response: We changed “function” to “functions” in the revised article.
Figure 2 – remove the black box over Fig. 2C
Response: We removed the black box in the Fig. 2.
Line 181 – should read AIR not AIRs
Response: We removed “s” from “AIRs”
Line 225 – an “introduced cDNA fragment” is a more accurate description than an “artificial fragment”
Response: Thank you for the nice comment. We used “introduced cDNA fragment” to replace “artificial fragment” in the revised article.
Line 253 – should read Influences not Influence
Response: Yes, we changed influence to influences in the revised manuscript.
Line 267 – should read resulted not result
Response: it could be better to delete “have” in this sentence. We have already revised this sentence in the revised manuscript.
Line 269 – do the authors mean detectable rather than performed?
Response: We have already amended this sentence in the revised manuscript.
Line 360 – remove s from introns
Response: We removed “s” from introns.
Line 433 – do the authors mean the second internode?
Response: Yes, we did amend in the revised article.
Line 443 – remove s from AIRs
Response: We changed AIRs to AIR.
Reviewer 2 Report
Major points:
Overall, I enjoyed reading this manuscript. It is well written with only a few minor typographical / English expression errors (see below under Minor points for details). However, there are a few areas that require further detail and/or clarification.
Line 95 – can the number of T2 plants (x of y plants with the drooping and fragile phenotype) be provided?
Line 117 – the authors described the vascular bundles of the df mutant culm as shorter but that they have no difference in width (Fig. S1). Is this due to less cells being within the vascular bundle (as is observed in the parenchyma layer) and/or a reduction in expansion of these cells? Please mention that there is no effect on the width of the vascular bundle within the body of the text.
Figure 2 – Where do the cells shown in Fig. 2F sit in relation to Fig. 2D? Some labels should be added to Fig.’s 2D and 2F to indicate cell types and contents, respectively. Are the authors sure that the cell walls of the df mutant are electron-dense? The cellular contents of the displayed cells have condensed contents suggesting that the fixation of the plant material wasn’t optimal, hence, the differences may be artefactual.
Line 140-141, 143-144 – What is meant by the “abaxial end of the main vein”. The term abaxial refers to the bottom leaf blade surface. Do the authors mean the distal or proximal end of the main vein? Please clarify.
Figure 3 – please label sections with the different cell types
Table 1 – the values within this table do not appear in columns. Please provide units at the top of the table. Can the authors please comment on whether any (1,3;1,4)-β-glucan is present? Why are there only two callose values? To which samples do they belong to?
Line 196-197 – The 282 bp band appears to be absent from the DF plant rather than weakly expressed according to the included image. Can a more convincing image be provided? It is difficult to judge if the OsBGAL1 is highly expressed or not in this context. A change in wording to “more highly expressed” is suggested.
Line 223 – the number of T2 lines with a ~3:1 segregation ratio be provided? The reviewer recommends that the high temperature growth conditions be mentioned within the text in addition to the legend to Fig. 5 legend as these are the conditions under which a phenotype should be observed.
Figure 5 – the bands in Fig. 5B lanes 1-1 and 2-1 are difficult to see? Can a more convincing image be added instead? It would be useful to point out that lines 1-18, 1-19 and 2-8 and 2-9 are "nulls" for the introduced DF cDNA, at least in the legend.
Figure 6 – What are the black and white arrows within the green boxes? Is the length of the last green box indicative of the shorter 257 bp 3' UTR? It is unclear where the 2 polyA tail addition sites are located in this figure and in Figure S5B (refer to Line 287). Presumably the location of qRT1 and qRT2 primer pairs detect the different transcripts but it is not clear from the diagram. Please clarify.
Line 318 – insert after be, "one or a combination of the following possibilities."
Line 360 – add in the citation to reference 38
Line 375 – list the name of the DNA sequencing company
Line 388 – please provide a locus number for the rice RUB1 gene
Line 399 – list the type of nylon and the manusfacturer
Line 411, 416, 423 – please add more details of the drying and embedding process
Line 306-325 - The early part of the Discussion focusses on the possible mechanism/s by which the introns of the premRNA transcript of DF aren’t spliced. Was genetic complementation also attempted with DF gDNA to confirm the importance of the native DF locus context to this nonsplicing?
Supplemental material
Line 40 – Replace “Statistic analysis of the toppest three leaves length” with “Statistical analysis of the length of the top three leaves”
Line 42 – what do the arrows indicate? Please add cell labels to the image
Line 46-47 – What do the authors mean by “The experiment comes from triplicate for 4 independent biological repeats.” Are the images within Fig. S4 taken from one biological replicate and is representative of the others? Could the main anatomical features of the sections be labelled?
Figure S5 – it is unclear where the two polyadenylation sites are in the DF locus. Please mark
The major omission to the manuscript is the lack of description as to the consequence of nonspliced introns within the DF transcript at the end of the Results section. What is the impact on the DF protein when the introns aren't spliced out? Is the mature protein truncated due to a premature stop codon or does it encode a different C-terminus compared to the wild type protein? Is it predicted to still be functionally active? An understanding of the impact on the pool of GPI-anchored proteins would be helpful to gain an understanding of the phenotype of this mutant. This would then set the biological context of these findings within the discussion. Can the authors provide insight as to why so many different cell wall components differ between the RCCR1i lines and the df mutant?
Minor points:
Line 34 – describe EtNP in full prior to using this abbreviation.
Line 63 – should read root hairs not root hair
Line 72 – should read functions not function
Figure 2 – remove the black box over Fig. 2C
Line 181 – should read AIR not AIRs
Line 225 – an “introduced cDNA fragment” is a more accurate description than an “artificial fragment”
Line 253 – should read Influences not Influence
Line 267 – should read resulted not result
Line 269 – do the authors mean detectable rather than performed?
Line 360 – remove s from introns
Line 433 – do the authors mean the second internode?
Line 443 – remove s from AIRs
Author Response
Comments and Suggestions for Authors
Reviewer2
Overall, I enjoyed reading this manuscript. It is well written with only a few minor typographical / English expression errors (see below under Minor points for details). However, there are a few areas that require further detail and/or clarification.
Line 95 – can the number of T2 plants (x of y plants with the drooping and fragile phenotype) be provided?
Response: Thank you for your comments.1/4 plants show the drooping and fragile phenotype, and we added the sentence “eight of thirty plants in the T2 generation showed drooping and fragile phenotype” in the revised manuscript with “Track Changes”.
Line 117 – the authors described the vascular bundles of the df mutant culm as shorter but that they have no difference in width (Fig. S1). Is this due to less cells being within the vascular bundle (as is observed in the parenchyma layer) and/or a reduction in expansion of these cells? Please mention that there is no effect on the width of the vascular bundle within the body of the text.
Response: Thank you for your comments. In FigS1, df culm showed the shorter length but normal width in vascular bundles, therefore, we said the vascular bundles of the df mutant culm was shorter. For the avoidance of confusion, we revised the sentence as “the length of the vascular bundle is shorter’ in the revised manuscript with “Track Changes”.
Figure 2 – Where do the cells shown in Fig. 2F sit in relation to Fig. 2D? Some labels should be added to Fig.’s 2D and 2F to indicate cell types and contents, respectively. Are the authors sure that the cell walls of the df mutant are electron-dense? The cellular contents of the displayed cells have condensed contents suggesting that the fixation of the plant material wasn’t optimal, hence, the differences may be artefactual.
Response: Thank you for your comments. Yes, the electron-dense could be caused in the process of the fixation. But here we would like to show the different layers in the cell wall that was indicated with arrows in Fig 2F. Now we changed the sentence as “Whilst showed a clearly layered structure with different electron-dense materials in df cell walls” in the revised article with “Track Changes”.
Line 140-141, 143-144 – What is meant by the “abaxial end of the main vein”. The term abaxial refers to the bottom leaf blade surface. Do the authors mean the distal or proximal end of the main vein? Please clarify.
Response: Thank you so much for your suggestion. Yes, it should be the distal/proximal end of the main vein. We revised the article accordingly.
Figure 3 – please label sections with the different cell types
Response: Thank you for your comments. Figure 3C and 3D are the close-up of sections from A and B, so we labeled the sections and also the different cell types in the revised Figure 3 and its legend as well.
Table 1 – the values within this table do not appear in columns. Please provide units at the top of the table. Can the authors please comment on whether any (1,3;1,4)-β-glucan is present? Why are there only two callose values? To which samples do they belong to?
Response: We corrected the format of Table 1, added the units at the top of the table based on your nice comments. In the revised Table, we thought it’s better for us to put the callose value in the section of “2.5. Cloning of the Gene”, instead of Table 1. Therefore, the description in the revised manuscript is as following: Mutations in the AtGPI8 gene cause the atgpi8-1 mutant form stomatal clustering and accumulate higher levels of callose in Arabidopsis seedlings [12]. We determined the callose (1,3-β-glucan) contents in RCCR1i and df leaves at the four-leaf stage, and observed that df leaves had a significantly higher callose content (43.58±0.24 μg g-1 fresh weight (FW)) than RCCR1i leaves (21.26±0.37 μg g-1FW). But no stomatal clustering was observed in df leaf blades as described above (Supplementary Figure S3B).
Line 196-197 – The 282 bp band appears to be absent from the DF plant rather than weakly expressed according to the included image. Can a more convincing image be provided? It is difficult to judge if the OsBGAL1 is highly expressed or not in this context. A change in wording to “more highly expressed” is suggested.
Response: We used an improved image in Figure 4 in the revised article and changed the sentence based on your reasonable suggestion.
Line 223 – the number of T2 lines with a ~3:1 segregation ratio be provided? The reviewer recommends that the high temperature growth conditions be mentioned within the text in addition to the legend to Fig. 5 legend as these are the conditions under which a phenotype should be observed.
Response: We added the separation ratio and the description of phenotype in the revised manuscript according to your nice comments. And we also added the sentence “The ratio of normal phenotype to df mutant phenotype stands at 17: 5 and 7: 2 in two independent lines, respectively” in the legend of Figure 5.
Figure 5 – the bands in Fig. 5B lanes 1-1 and 2-1 are difficult to see? Can a more convincing image be added instead? It would be useful to point out that lines 1-18, 1-19 and 2-8 and 2-9 are "nulls" for the introduced DF cDNA, at least in the legend.
Response: Fig. 5B was replaced by an improved image and the missing band in lines 1-18, 1-19 and 2-8 and 2-9 was indicated in the Figure 5.
Figure 6 – What are the black and white arrows within the green boxes? Is the length of the last green box indicative of the shorter 257 bp 3' UTR? It is unclear where the 2 polyA tail addition sites are located in this figure and in Figure S5B (refer to Line 287). Presumably the location of qRT1 and qRT2 primer pairs detect the different transcripts but it is not clear from the diagram. Please clarify.
Response: We re-organized the Figure 6 in the revised manuscript and made all the data more clear according to your nice suggestions. PolyA tail was indicated in Figure S5B.
Line 318 – insert after be, "one or a combination of the following possibilities."
Line 360 – add in the citation to reference 38
Response: We added the reference in the corresponding position and reorganized the order of references as well.
Line 375 – list the name of the DNA sequencing company
Response: We added the name of the company in the revised paper.
Line 388 – please provide a locus number for the rice RUB1 gene
Response: We added the locus number LOC_Os06g46770 in the revised paper.
Line 399 – list the type of nylon and the manusfacturer
Response: We added the name of the company in the revised paper.
Line 411, 416, and 423 – please add more details of the drying and embedding process
Response: We added the details about the drying and embedding process in the revised manuscript as “Subsequently, samples were dehydrated successively with 30% (4ºC, 20 min), 50% (4ºC, 20 min), 70% (4ºC, overnight), 80% (RT, 15 min), 90% (RT, 15 min), 100% alcohol (RT, 30 min for twice) and finally propylene oxide (RT, 30 min for twice). After dehydration, samples were treated by different ratio of propylene oxide and epoxy resin, and then embedded in in pure epoxy resin.”
Line 306-325 - The early part of the Discussion focusses on the possible mechanism/s by which the introns of the premRNA transcript of DF aren’t spliced. Was genetic complementation also attempted with DF gDNA to confirm the importance of the native DF locus context to this nonsplicing?
Response: Thank you for your suggestions. Here we used the full length of cDNA (including 5’UTR, ORF and 3’UTR) to make genetic complementation in df background in our research.
Supplemental material
Line 40 – Replace “Statistic analysis of the toppest three leaves length” with “Statistical analysis of the length of the top three leaves”
Response: We replaced the sentence based on your nice comment with “Track Changes”.
Line 42 – what do the arrows indicate? Please add cell labels to the image
Response: Arrows indicated a column of stomatal cells that do not differ between df and RCCR1i plants. We added this sentence in the legend of Fig. S3.
Line 46-47 – What do the authors mean by “The experiment comes from triplicate for 4 independent biological repeats.” Are the images within Fig. S4 taken from one biological replicate and is representative of the others? Could the main anatomical features of the sections be labelled?
Response: We repeated this experiments for three times and used 4 independent plants for each experiment, since we needed to exclude that the shrinkage structure in df mutant was caused by the fixation of plant materials. Now we reorganized the sentence in the legend of Fig.S4 and labelled the main difference between df and RCCR1i.
Figure S5 – it is unclear where the two polyadenylation sites are in the DF locus. Please mark.
Response: We marked the polyadenylation sites in the image as 3907 and 4409, and added the detailed description about these sites in the legend of Figure S5 accordingly.
The major omission to the manuscript is the lack of description as to the consequence of nonspliced introns within the DF transcript at the end of the Results section. What is the impact on the DF protein when the introns aren't spliced out? Is the mature protein truncated due to a premature stop codon or does it encode a different C-terminus compared to the wild type protein? Is it predicted to still be functionally active? An understanding of the impact on the pool of GPI-anchored proteins would be helpful to gain an understanding of the phenotype of this mutant. This would then set the biological context of these findings within the discussion. Can the authors provide insight as to why so many different cell wall components differ between the RCCR1i lines and the df mutant?
Response: Thank you for your nice comments. The mature protein truncated is due to a premature stop codon in the transcript. It lacks the main active domain and therefore the functions of OsGPI8. We added some sentences in the “2.5. Cloning of the Gene” to describe the consequence of the retained introns in DF transcript as “While the intron retention in transcripts of the OsGPI8 gene results in the premature stop codon and the encoded truncated protein lacking the main active domain. Therefore, the retained introns in DF transcript makes OsGPI8 lost functions.”
We annotated 481 GPI-anchored proteins from rice (Supplementary Table 2) and discussed the pool in the “Discussion”. From the annotation in Supplementary Table 2, we can see that GPI-anchored proteins play multiple roles in the Cell wall deposition and therefore the cell wall components in df are remarkably different from the RCCR1i.
Minor points:
Line 34 – describe EtNP in full prior to using this abbreviation.
Response: The abbreviation list has been completed in the revised version according to reviewers’ nice comments.
Line 63 – should read root hairs not root hair
Response: We changed “root hair” to root hairs in the revised article.
Line 72 – should read functions not function
Response: We changed “function” to “functions” in the revised article.
Figure 2 – remove the black box over Fig. 2C
Response: We removed the black box in the Fig. 2.
Line 181 – should read AIR not AIRs
Response: We removed “s” from “AIRs”
Line 225 – an “introduced cDNA fragment” is a more accurate description than an “artificial fragment”
Response: Thank you for the nice comment. We used “introduced cDNA fragment” to replace “artificial fragment” in the revised article.
Line 253 – should read Influences not Influence
Response: Yes, we changed influence to influences in the revised manuscript.
Line 267 – should read resulted not result
Response: it could be better to delete “have” in this sentence. We have already revised this sentence in the revised manuscript.
Line 269 – do the authors mean detectable rather than performed?
Response: We have already amended this sentence in the revised manuscript.
Line 360 – remove s from introns
Response: We removed “s” from introns.
Line 433 – do the authors mean the second internode?
Response: Yes, we did amend in the revised article.
Line 443 – remove s from AIRs
Response: We changed AIRs to AIR.
Reviewer1
In my opinion, the aim of the work is not fully clear. This point should be highlighted in comparison with what has been achieved in previous studies. Additionally, no functional studies or mutational approaches were performed. These points should be also clarified with explanation.
Response: Thank you for your nice comments. We highlighted the significance of our results in the Introduction and made a detailed comparison in the discussion as well in the revised manuscript. Furthermore, we added a supplementary data about the annotation of all the GPI-APs from rice plant, which helps us to understand the significance of our results and also the function of GPI-Aps better.
Introduction
The aim of the paper should be more clarified and highlighted in this section.
Response: Thank you for your constructive comments. We rewrote the last paragraph of introduction and clarified the significance of our results. New paragraph was written as following: “In the present study, we further elucidated the biology function of GPI-APs in rice plants using a T-DNA insertional mutant that shows a drooping and fragile (df) phenotype. Tail-PCR and DNA sequencing indicated that the T-DNA was inserted downstream of the stop codon of OsGPI8, the ortholog of the Arabidopsis GPI8 gene. Insertion of the T-DNA causes defective intron splicing of the OsGPI8 pre-mRNA, especially under higher temperature conditions. Intron retention in transcripts of the OsGPI8 gene results in the premature stop codon. Results from anatomical analysis showed that the insertion mutation led to a decreased cell number in the culm and a large change in cell wall structure. Reduced cellulose and lignin contents were produced in the mutant culms and leaves correspondingly. Our results provide a new insight into the function of GPI-anchored proteins in plant cell division in rice plants”.
Results
Figure 1: Do you mean SD or mean ± SD? Please clarify this point.
Response: Thank you for the comments. SD should be ± SD in our manuscript and we have already clarified this point.
2.2. Changes in Morphological and Anatomical Structures of Culms
Figure 2: All figures of statistical analyses are not understandable, where the relation between X-axis and Y-axis is not described below the figure.
Response: Thank you for the comments. We have already added detailed descriptions of X-axis and Y-axis below figures in the revised manuscript with "Track Changes".
2.3. Changes in the Morphological and Anatomical Structures of Leaves
Figure 3. The description below the figure is very confusing and difficult to understand. The figure should be reorganized with more clarification.
Response: Thank you for the comments. We have already reorganized the description in figure 3 in the revised manuscript with "Track Changes".
2.4. Differences in Cell Wall Composition
In my opinion, the authors did not test all the cell wall composition, where they focused mainly on the sugar content (carbohydrate). This point should be clarified, and the importance of the chosen components should be also clarified in the text.
Response: Thank you for your comments. Cellulose, hemi-cellulose and lignin that are related to mechanical force were determined. We used xylose content to represent hemi-cellulose in the previous version. Now we added the content of hemi-cellulose in the revised manuscript and all the revised parts were used "Track Changes".
Table 1. The data in Table 1 are not clear and the table should be reorganized.
Response: Thank you for your comments. We have already reorganized Table 1 in the revised manuscript.
Discussion
This section needs more improvement, where the discussion of the obtained results are poorly discussed and should be rationalized with more adequate previous studies.
Response: Thank you for your nice comments. We reorganized the Discussion. And we added a supplementary data about the annotations of all the GPI-APs from rice plant, which helps us to understand the significance of our results and also the function of GPI-APs better.
Materials and methods
4.3. RNA Isolation and Expression Analysis
Lines 388-389. Based on which criteria the primers have been chosen? This point needs clarification in the text.
Response: Rice polyubiquitin1 (OsRub1) is expressed constitutively in rice plant. We added the Locus number and also reference in the revised manuscript.
4.8. Cell Wall Component Measurement
I would like to see the all chromatogram data of the analyzed silylated sugar by GC-MS. This could be added in the supplementary file.
Response: We only detected several specific cell wall components with the method we mentioned in the article, and all the data we obtained have been already presented in the manuscript. Now we corrected the Method and added the reference in the revised manuscript. What we rewrote is “The destarched AIR are further hydrolyzed into monosaccharides by trifluoroacetic acid (TFA) and reduced with sodium borohydride. The alditol acetate derivatives produced by acetic anhydride treatment are subjected to GC-MS according to Zhang and Zhou (2017) [32].”
Finally, I recommend the authors to check the whole paper for grammatical and typing errors as well as the list of abbreviations. For instance, some abbreviations are not listed such as SD (standard deviation).
Response: We checked and revised the manuscript carefully based on your nice comments. And we also reorganized the list of abbreviations in the revised article.
Round 2
Reviewer 1 Report
The manuscript has been significantly improved.